# Using de-identified electronic health records to research mental health supported housing services: A feasibility study

**Christian Dalton-Locke**[1]*, **Johan H. Thygesen**[1,2], **Nomi Werbeloff**[1,2], **David Osborn**[1,2], **Helen Killaspy**[1,2]

**1** Division of Psychiatry, University College London, London, United Kingdom, **2** Camden and Islington NHS Foundation Trust, London, United Kingdom

* c.dalton-locke@ucl.ac.uk

**Data Availability Statement:** The data used in this work has been obtained from the CRIS tool, a system which has been implemented at CIFT. It provides authorised researchers with regulated

## Abstract

### Background

Mental health supported housing services are a key component in the rehabilitation of people with severe and complex needs. They are implemented widely in the UK and other deinstitutionalised countries but there have been few empirical studies of their effectiveness due to the logistic challenges and costs of standard research methods. The Clinical Record Interactive Search (CRIS) tool, developed to de-identify and interrogate routinely recorded electronic health records, may provide an alternative to evaluate supported housing services.

### Methods

The feasibility of using the Camden and Islington NHS Foundation Trust CRIS database to identify a sample of users of mental health supported accommodation services. Two approaches to data interrogation and case identification were compared; using structured fields indicating individual's accommodation status, and iterative development of free text searches of clinical notes referencing supported housing. The data used were recorded over a 10-year-period (01-January-2008 to 31-December-2017).

### Results

Both approaches were carried out by one full-time researcher over four weeks (150 hours). Two structured fields indicating accommodation status were found, 2,140 individuals had a value in at least one of the fields representative of supported accommodation. The free text search of clinical notes returned 21,103 records pertaining to 1,105 individuals. A manual review of 10% of the notes indicated an estimated 733 of these individuals had used a supported housing service, a positive predictive value of 66.4%. Over two-thirds of the individuals returned in the free text search (768/1,105, 69.5%) were identified via the structured fields approach. Although the estimated positive predictive value was relatively high, a substantial proportion of the individuals appearing only in the free text search (337/1,105, 30.5%) are likely to be false positives.

access to de-identified information extracted from patient electronic health records. CRIS is governed by a strict information governance scheme which forbids anyone except for authorised researchers from accessing its records (see 'Ethics approval and consent to participate'). For further details and requests to access the SQL code used in the present study, including how it was developed, please contact: researchdatabase@candi.nhs.uk.

**Funding:** Christian Dalton-Locke is funded by a 1 +3 Economic and Social Research Council (ESRC) PhD studentship. Prof Killaspy is supported by the UCLH NIHR Biomedical Research Centre. Prof Osborn is supported by the UCLH NIHR Biomedical Research Centre and he was also in part supported by the National Institute for Health Research (NIHR) Collaboration for Leadership in Applied Health Research and Care (CLAHRC) North Thames at Bart's Health NHS Trust.

**Competing interests:** The authors have declared that no competing interests exist.

## Conclusions

It is feasible and requires minimal resources to use de-identified electronic health record search tools to identify large samples of users of mental health supported housing using structured and free text fields. Further work is needed to establish the availability and completion of variables relevant to specific clinical research questions in order to fully assess the utility of electronic health records in evaluating the effectiveness of these services.

## Background

Specialist mental health supported accommodation services are a key component of community based mental health service in many countries and support people with complex and longer term mental health problems [1]. It is estimated that around 60,000 people use these services in the UK at any time [2, 3]. Three main types have been described: residential care homes provide long term, high support for the most disabled group (communal facilities, staffed 24 hours, support with all activities of daily living (ADL) including meals, self-care, cleaning, budgeting, medication management, etc.); supported housing services aim to help individuals gain ADL and vocational skills so they can move-on to more independent accommodation (self-contained or shared tenancies, staffed up to 24 hours a day, time-limited); floating outreach services provide visiting support for a few hours a week to individuals living in a permanent, self-contained, individual tenancy with the aim of reducing the hours of support over time to zero. Recently, a new typology for supported accommodation has been developed, the Simple Taxonomy–Supported Accommodation (STAX-SA) [4]. Residential care maps to STAX-SA Type 1, supported housing services map to Types 2 and 3, and floating outreach services map to Types 4 and 5.

A recent national programme of research into mental health supported accommodation services across England (the 'QuEST study'), which included a national survey [1] and a cohort study following 619 service users from 87 services over 30 months [5], has demonstrated it is possible to conduct high-quality studies in this area using traditional research methods (face to face interviews with participants). However, it also demonstrated that trials in this area are not feasible [6] and that studies using these research methods in this field are lengthy to conduct and expensive. It took a team of three full-time researchers over five years to complete data collection, with a total programme cost of around £2 million.

Furthermore, a recent systematic review found there was a lack of empirical evidence, and of the studies that have been conducted there was wide heterogeneity in the terminology used to describe services and the outcomes assessed, such that the evidence could not be synthesised to inform practice or service planning [7]. Therefore, more high quality studies using alternative and more resource efficient research methods are required.

The use of routinely collected healthcare records may provide an alternative approach in assessing outcomes and effectiveness of these services. These records contain detailed, longitudinal, clinical data and, in recent years, their use in research has been facilitated by the switch from paper to digital records, encouraged by the UK Government's aim to create a 'paperless NHS' [8]. Researchers no longer need to interrogate handwritten paper records available only at the sites they are physically stored but can instead access electronic health records (EHR) on a computer/device with the appropriate access and information governance permissions.

However, EHR contain confidential personal data, and are only accessible to researchers if they have the informed consent of the individual, requiring participant recruitment and the

appropriate permissions from the relevant healthcare organisation(s) (and incurring much of the costs associated with traditional research). To address this issue, the Biomedical Research Centre at South London and Maudsley NHS Foundation Trust (SLaM) developed a tool to extract and de-identify data from EHR, the Clinical Record Interactive Search (CRIS) system [9–11]. It locates 'Patient Identifiers' as stipulated by the Caldicott Code on Confidentiality, and de-identifies them by removing or replacing the identifiable data (such as the individual's name with 'ZZZZZ'). The tool can interrogate both structured and free text fields. Structured fields hold a range of demographic and clinical information (such as sex, date of birth, diagnosis, etc.) completed by selecting from a list of options (for example, male/female) or by using a specific format (date). Although structured fields provide data that easily lends itself to quantitative analysis, their utility in research is limited by the structured fields that are available and their poor completion rates owing to the preference of clinicians to record data in natural language [12]. Free text fields include any entered text and typically comprise clinical notes and uploaded documents. They represent an estimated 60–70% of the data in EHR [10, 12] and are a rich source of clinical information, ranging from psychiatric assessments to logs of clinical appointments and referrals.

CRIS has been shown to be very accurate in its de-identification [10] and has been approved for use in mental health research, without requiring individuals' informed consent [9]. Dozens of studies using CRIS with SLaM's EHR have been published on a number of topics including the characteristics of trafficked adults and children with severe mental illness [13], and the outcomes for users of mother and baby units [14]. However, de-identified EHR have not yet been used to research mental health supported accommodation. It is currently unknown whether it is feasible to identify a sample of individuals who use supported accommodation services using CRIS. In the UK, these services are mainly provided by charities or housing associations and not the NHS. However, these services are an essential component of care for people with complex and enduring mental health problems, most of whom are also using NHS mental health services and therefore will have EHR [1]. There is potential then to employ CRIS as a research tool to evaluate supported accommodation services.

In 2013, the CRIS tool was deployed in four additional Trusts in the UK, including Camden and Islington NHS Foundation Trust (CIFT). We undertook a recent audit of mental health supported accommodation services in Camden and Islington which identified 57 services providing 783 places at any one time (12 residential care services providing 230 places, 34 supported housing services providing 390 places and 9 floating outreach services providing 163 places).

We aimed to assess the feasibility of using CRIS to derive a sample of past and present users of mental health supported accommodation services from CIFT's EHR by identifying any relevant structured fields and developing a search of free text records.

## Methods

### Setting

Camden and Islington are both inner London boroughs with a combined population of approximately 470,000. They have a lower proportion of older adults compared to the rest of England (aged 65+ in Camden 11.7%, Islington 8.7%, England 17.7%), more people from Black, Asian and Ethnic minority groups (Camden 33.7%, Islington 32.0%, England 14.5%) [15] and a higher prevalence of adults with psychotic disorders (0.6% Camden, 0.7% Islington, 0.4% England) [16]. Of the 326 local authority areas in England, Camden ranks 74th and Islington 14th as the most deprived [17]. Across both boroughs, supported housing services are provided by several different voluntary organisations and housing associations.

CIFT provides inpatient and community mental health services for both boroughs, including general adult, rehabilitation, substance misuse, learning disability and homelessness services, and holds statutory care co-ordination responsibility for people with severe mental health problems subject to the Care Programme Approach (CPA). The Trust's records were paper based until 2008 when an EHR system was installed. At the end of 2017 there were 126,769 individuals with records in this system [18]. The CRIS tool has been used to de-identify these records making them available for research.

## Search approach

Using CRIS, we explored the potential utility of two approaches to obtain a sample of de-identified individuals who have used a supported housing service: i) using structured fields relevant to the individual's accommodation; and ii) developing a free text search of clinical notes. We also compared the two approaches to see if they identified the same individuals. Finally, we investigated whether it was possible to describe the sample in terms of their clinical characteristics and sociodemographics using structured fields in the CRIS data set, and compare this to a national survey carried out in 2014 which included 619 service users from 87 supported accommodation services. Records between 1 January 2008 and 31 December 2017 were examined. The searches were developed and conducted by CD-L, who has experience as a clinician and researcher in the field of supported accommodation. To assess the potential resource effectiveness of this approach compared to the amount of time and multiple researchers normally required in primary research that requires participant recruitment and research interviews, we limited the available time to carry out the searches to 150 hours (one full-time researcher for four weeks).

## Structured fields search for individuals using supported accommodation services

Two structured fields relevant to an individual's accommodation were identified in the sections of the EHR where the clinician is expected to record and update CPA meeting outcomes ('cpa_accommodation_desc') and demographic details ('accommodation_status_desc'). Both these fields had values (options for the clinician to enter) representative of mental health supported accommodation services. We included values representative of all types of supported accommodation and not just supported housing services because of the heterogeneity in the terminology used to describe supported housing services, and the likelihood that single values are used by different clinicians to record different types of service. The included values were: 'Supported accommodation', 'Supported lodgings', 'Supported group home', 'Mental Health Registered Care Home', and 'Other accommodation with mental health care and support'. For a full list of all the response options (values) available to clinicians for both structured fields, see S1 Table. All entries using either of these fields are stored, so it is possible to identify previous as well as current accommodation status.

## Free text search of de-identified clinical notes

The flow diagram shown in Fig 1 illustrates the iterative process for the free text search of clinical notes. First, a list of all supported housing services in the area was generated, based on a previous audit we carried out and verified with senior managers of local mental health supported accommodation services. The final list comprised 35 services. A series of single service searches were developed for each service based on the name of the service. As there were four pairs of services with similar names, 31 single service searches were developed, before combining these in an 'all service search'. The search started at a simplistic level, by using the most

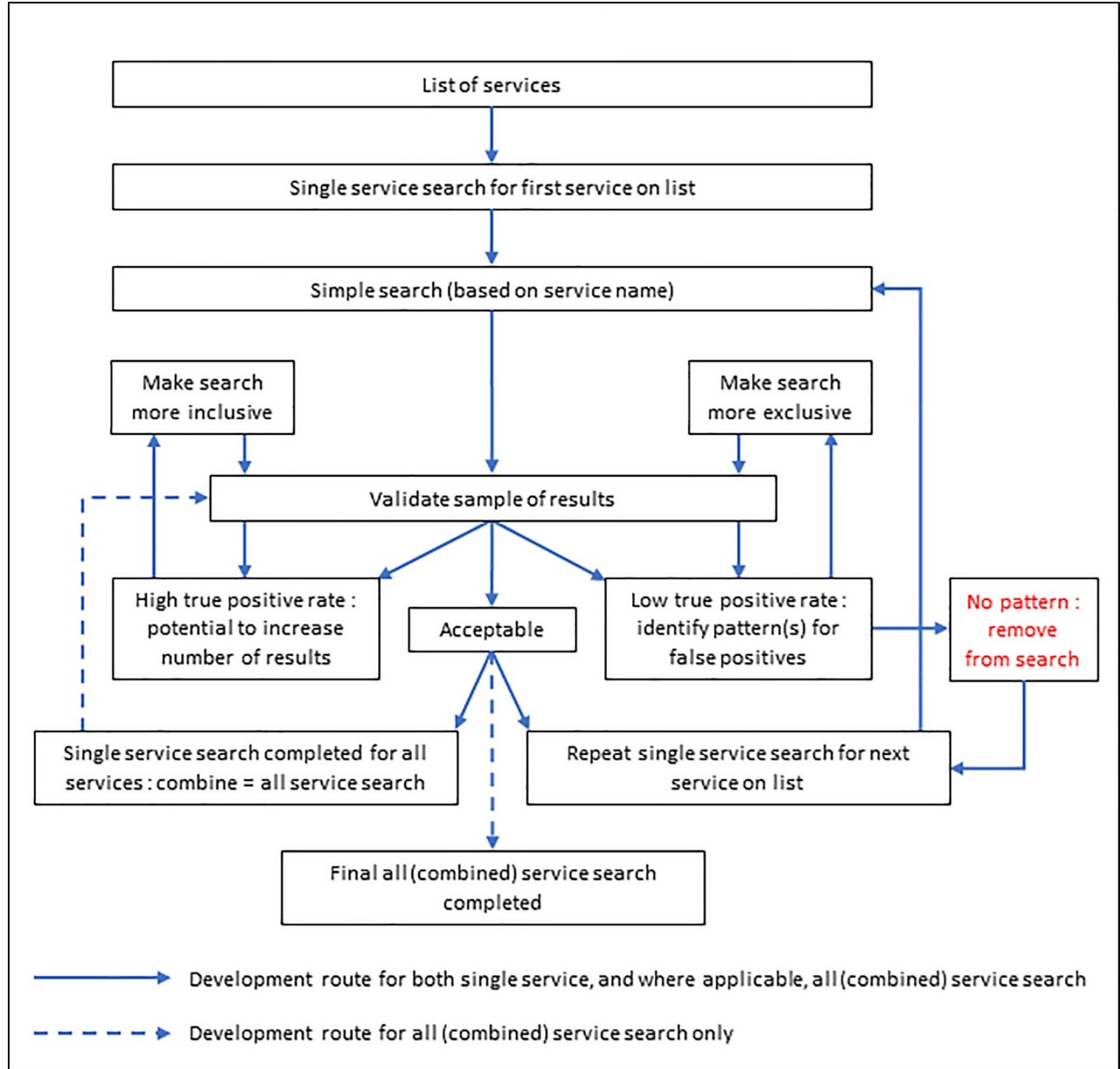

**Fig 1. Flow diagram of free text search development.**

distinctive word from the name of the service so that all clinical notes with a mention of this word were returned. Returned results included the unique identification number randomly assigned by CRIS to each individual on the database, the text of clinical notes that contained the search term and the date the note was recorded. Results were ordered by identification number, which is randomly assigned, and note date, to facilitate manual review. Refinement of the search process was iterative and based on a manual notes review, so that the returned results could be improved in terms of the number of notes they contained, the number of individuals they pertained to and the ratio of true positives to total positives, i.e. the positive predictive value.

The manual notes review consisted of clinical notes pertaining to the first 10% of individuals listed. If it was clear from the individual's note(s) that they had previously used or were currently using a supported housing service, the individual (not individual notes) was designated as a true positive. A typical example would be a note documenting a clinician's visit to a service to see the individual. An individual was assigned as false positive if the notes pertaining to that individual were not actually referring to a supported housing service or if a service was being referred to but it was unclear if the individual had ever actually used the service.

Reasons for false positives were noted and if any pattern(s) emerged, used to improve the search term. For example, a search for the fictitious service 'Forward View' would initially be based on the search term FORWARD, which would return clinical notes including mentions of Forward View but also any other mention of the word 'forward'. The search could then be improved by further specifying the search term to FORWARD V or adding terms so that results could not contain the terms FORWARD THINKING or FORWARD PLANNING. Patterns to false positives were not limited to text included in the clinical note but could also include patterns such as the number of notes returned per individual. For example, if a false positive were more likely than a true positive to only have a single note returned by the search, then the search could be refined to only include individuals which have more than one note pertaining to them.

This process was repeated until there was no longer a consistent pattern to the false positives and the positive predictive value was acceptable, i.e. over 25%. If the positive predictive value was high, i.e. over 75%, the search was revised to see if a higher number of returns could be achieved whilst maintaining a high positive predictive value. The process was therefore a matter of attempting to achieve the optimal balance between specificity (not over inclusive and lacking in accuracy) and sensitivity (not over exclusive and lacking in sample size). If a pattern to the false positives did not appear and the positive predictive value was not acceptable, development of this single service search was stopped and it was not included in the all service search.

This procedure was repeated for each supported housing service. The finalised searches for each service were then combined to produce an 'all service search'. This search went through the same procedure of development and refinement as the single service searches, which eventually produced a 'final all service search' from which the estimated positive predictive value was determined.

## Ethics statement

Researchers who wish to use the CIFT Research Database (CIFT records de-identified by CRIS) are required to have an honorary contract or letter of access with the Trust, and submit the CRIS Project Application form to the Oversight Committee', detailing the proposed study including the parameters of searches. The form is available here: http://www.candi.nhs.uk/health-professionals/research/ci-research-database/researchers-and-clinicians. An application for the present study was submitted and approved, and the lead researcher (CDL) has an honorary contract with CIFT. All studies using the CIFT Research Database have been granted ethical approval by the NRES Committee East of England—Cambridge Central (14/EE/0177).

## Results

### Structured fields search

Values representative of mental health supported accommodation in the CPA and demographics accommodation status structured fields were recorded for a total of 1,635 and 882 individuals, respectively. A large majority of the total 126,769 individuals in the database did

not have any record for either of these structured fields, and both fields are intended to record any type of accommodation. There are a total of 59,408 records using the CPA accommodation field and 65,065 records using the demographics accommodation field; in both instances multiple records can pertain to the same individual. See S1 Table for a full list of response options for each structured field and how these options were grouped, and S2 and S3 Tables to see how many individuals are in each group.

### Free text search of clinical notes

Table 1 illustrates the development of the free text search of clinical notes to identify people who had used a supported housing service. Of the 31 single service searches, 28 attained acceptable positive predictive values, the remaining three were removed and not included in the all service search. Half [14] of the single service searches had an acceptable positive predictive value after the first search, the most iterations required to develop an acceptable single service search was 9 (single service search 13).

The final all service search returned a total of 21,103 de-identified clinical notes pertaining to 1,105 individuals. Notes for 116 individuals (10.5%) were reviewed with a positive predictive value of 77/116 (66.4%). Extrapolating this rate to the remainder of the results produced an estimated positive predictive value of 733/1,105 (66.4%).

In the initial all services search, one of the key differences between true positive and false positive individuals was that false positives were much more likely to return with only a single clinical note for that individual, for true positives most often multiple notes would match. Therefore, a condition was added to the search whereby individuals were removed from the results if they only had a single note matching the search term. This largely explains the reduction in the number of individuals relative to the number of clinical notes between the 1$^{st}$ search (1,822 individuals and 23,501 notes) and the 1$^{st}$ iteration of the all service search (1,076 individuals and 22,755 notes, a reduction of 746 individuals and also a reduction of 746 notes). This was the only search condition applied that accounted for frequency patterns, and the only pattern/condition not based on the text content of notes. A full log of the search term development, including the identification of false positive patterns and the SQL search code, is archived on the CIFT CRIS Research Database and is available on request.

### Comparing the structured fields and free text search approach

Fig 2 shows how many individuals appeared in each of the three searches (the free text search of clinical notes and the two structured field searches), and the overlap between them. Of the 1,105 identified in the free text search, 739 (66.9%) were also identified in the CPA structured field, but only 249 (22.5%) also appeared in the demographics structured field. A total 768/1,105 (69.5%) of those identified in the free text search were also identified by one of the two structured field searches. The structured fields combined identified 2,140 unique individuals. All sources combined identified a sample of 2,477 unique individuals in total. Overall, 925 (37.2%) appeared in at least two of the searches and 220 (8.8%) appeared in all three. A total 337 individuals appeared only in the free text search.

### Sociodemographics of the structured fields and the clinical notes free text search samples

Table 2 shows the sociodemographics and diagnosis of the individuals identified from each search approach, extracted from structured fields within the EHR CRIS database, and from service users that participated in a national survey of supported accommodation carried out in 2014 [1]. Around two-thirds in each are male (59.3% - 66.8%), the mean age is between 41.7

**Table 1. Free text search development: The returned results for the first search, the first iteration and the final search for each service.**

| Search | | First search | | | First iteration | | | Final search | | | |
|---|---|---|---|---|---|---|---|---|---|---|---|
| | | Clinical notes | Individuals | Positive predictive value** | Clinical notes | Individuals | Positive predictive value** | Search or iteration no.* | Clinical notes | Individuals | Positive predictive value** |
| Single service search | 1 | 1582 | 266 | 58.3% | 179 | 73 | 81.8% | 1st | 1582 | 266 | 58.3% |
| | 2 | 1856 | 439 | 25.0% | 1677 | 340 | 33.3% | 5th | 233 | 88 | 66.6% |
| | 3 | 410 | 107 | 36.4% | 42 | 27 | . . . | 1st | 410 | 107 | 36.4% |
| | 4 | 824 | 108 | 50.0% | 13 | 8 | . . . | 1st | 824 | 108 | 50.0% |
| | 5 | 293 | 74 | 35.7% | 31 | 17 | . . . | 1st | 293 | 74 | 35.7% |
| | 6 | 749 | 245 | 0.0% | 711 | 217 | 7.1% | 4th | 256 | 47 | 85.7% |
| | 7 | 7256 | 1774 | 0.0% | 1851 | 938 | 0.0% | 5th | 410 | 117 | 100.0% |
| | 8 | 1979 | 842 | 0.0% | 595 | 262 | 11.1% | 6th | 268 | 108 | 100.0% |
| | 9 | 1039 | 284 | 14.3% | 982 | 241 | 6.5% | 4th | 96 | 36 | 100.0% |
| | 10 | 255 | 57 | 85.7% | 1003 | 173 | 85.7% | 2nd | 1003 | 173 | 85.7% |
| | 11 | 119 | 40 | 38.5% | 1181 | 100 | 66.6% | 2nd | 1181 | 100 | 66.6% |
| | 12 | 155 | 45 | 81.8% | 266 | 73 | 85.7% | 2nd | 266 | 73 | 85.7% |
| | 13 | 1472 | 637 | 0.0% | 965 | 464 | 0.0% | 10th | 160 | 84 | 62.5% |
| | 14 | 423 | 91 | 33.3% | 161 | 52 | 40.0% | 3rd | 54 | 33 | 41.7% |
| | 15 | 2923 | 191 | 63.6% | 441 | 86 | . . . | 1st | 2923 | 191 | 63.6% |
| | 16 | 1573 | 212 | 36.4% | . . . | . . . | . . . | 1st | 1573 | 212 | 36.4% |
| | 17 | 7110 | 305 | 50.0% | 6560 | 288 | . . . | 3rd | 4487 | 244 | 83.3% |
| | 18 | 2224 | 612 | 25.0% | 1018 | 250 | 17.6% | 3rd | 1005 | 240 | . . . |
| | 19 | 1758 | 897 | . . . | 1643 | 823 | 16.6% | 7th | 447 | 198 | 44.4% |
| | 20 | 431 | 73 | 44.4% | 28114 | . . . | . . . | 1st | 431 | 73 | 44.4% |
| | 21 | 752 | 344 | 0.0% | 81 | 17 | 15.4% | REMOVED | | | |
| | 22 | 217 | 74 | 71.4% | 217 | 75 | . . . | 2nd | 217 | 75 | . . . |
| | 23 | 798 | 124 | 33.3% | 33 | 20 | . . . | 1st | 798 | 124 | 33.3% |
| | 24 | 1107 | 315 | 0.0% | 1078 | 304 | 0.0% | REMOVED | | | |
| | 25 | 56 | 39 | . . . | 8151 | 1587 | . . . | 7th | 1046 | 345 | 28.6% |
| | 26 | 2111 | 175 | 66.7% | . . . | . . . | . . . | 1st | 2111 | 175 | 66.6% |
| | 27 | 319 | 101 | 66.7% | 267 | 117 | 33.3% | 1st | 319 | 101 | 66.7% |
| | 28 | 28592 | . . . | . . . | . . . | . . . | . . . | REMOVED | | | |
| | 29 | 1178 | 704 | . . . | 39 | . . . | . . . | 6th | 22 | 13 | 100.0% |
| | 30 | 150 | 32 | 37.5% | . . . | . . . | . . . | 1st | 150 | 32 | 37.5% |
| | 31 | 222 | 61 | 60.0% | . . . | . . . | . . . | 1st | 222 | 61 | 60.0% |
| All service search | 1 | 23501 | 1822 | . . . | 22755 | 1076 | 59.4% | 3rd | 21103 | 1105 | 66.4% |

Date range for search results: 01-Jan-2008 to 31-Dec-2017.

*1st = first search, 2nd = first iteration.

**Based on a manual review of a random 10% of identified individuals.

and 47.1, the proportions that are White range from 53.7% to 81%, most are single (66% - 83.9%), and the most frequently recorded diagnosis is schizophrenia or psychosis (39.0% - 63.6%). The greatest difference between the search approaches and the national survey was ethnicity, where 53.7% to 60.4% in the search approaches were White compared to 81% in the national survey. This reflects the greater proportion that are from Black, Asian and Ethnic minority groups in Camden and Islington compared to the rest of the country.

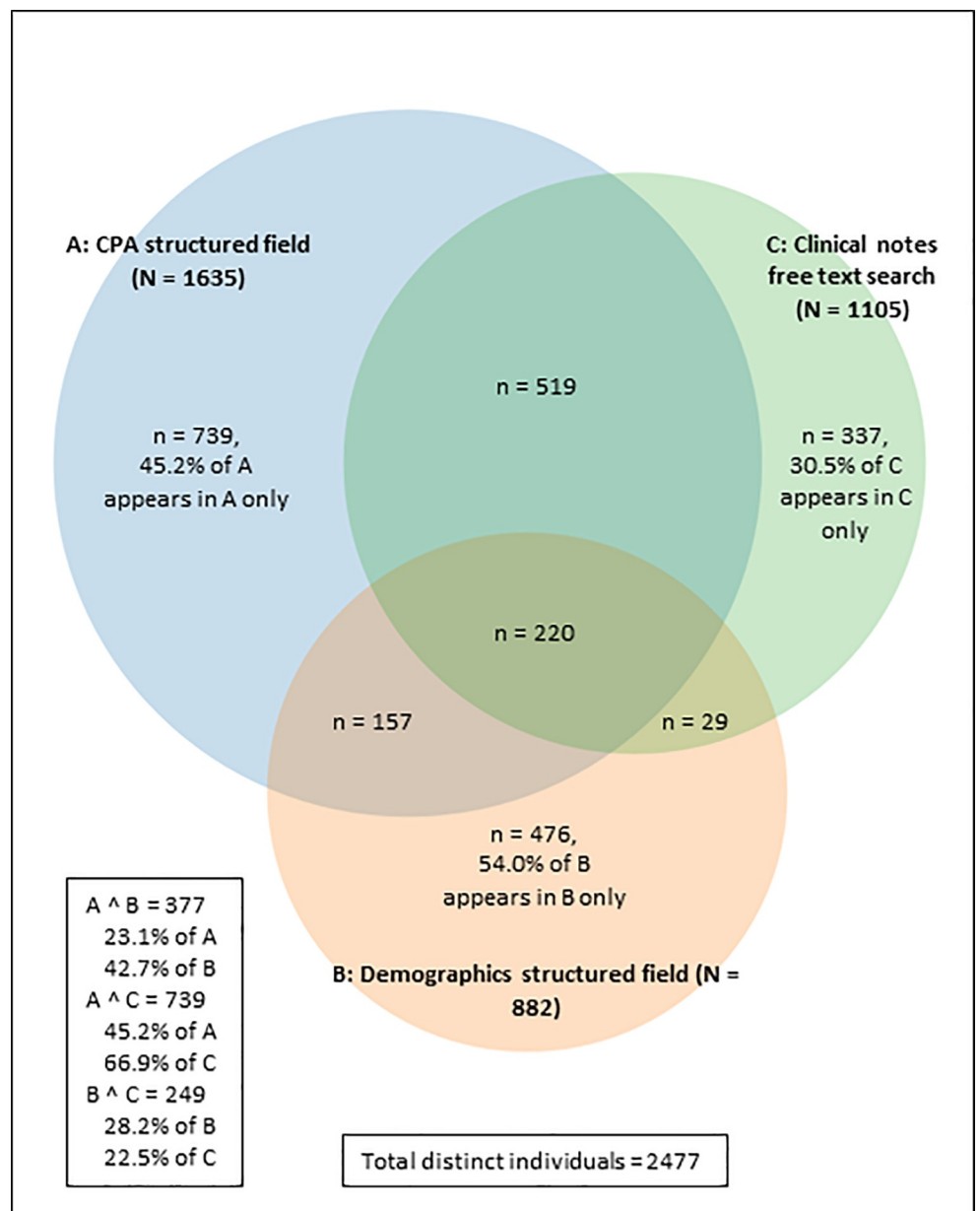

**Fig 2. A Venn diagram showing the overlap of individuals between the structured fields and free text search.**

## Discussion

To our knowledge, this is the first study to investigate the feasibility of using de-identified EHR within a confidentiality framework to identify a sample of mental health supported housing service users. We have shown that with very limited time and personnel resources relative to traditional research methods, it is possible to identify a large sample of de-identified people using these services and describe this sample in terms of their sociodemographics.

The overlap between the structured fields and the free text searches, which totalled 768 individuals, provides some degree of validation that this group had used a supported housing service. However, the utility of conducting the free text search in addition to the structured field searches needs to be considered. It is unlikely a clinician would take the time to complete the

**Table 2. Sociodemographics and diagnosis of the individuals identified by the different approaches, and from the national survey [1].**

| | | CPA structured field (N = 1635) | Demographics structured field (N = 882) | Clinical notes free text search (N = 1105) | National survey (N = 619)* |
|---|---|---|---|---|---|
| Sex—n (%) | Male | 1051 (64.3%) | 521 (59.1%) | 738 (66.8%) | 410 (66%) |
| | Unknown/Missing | 2 (0.1%) | 4 (0.5%) | 1 (0.1%) | 0 (0%) |
| Age† | Mean (SD) | 47.1 (16.3) | 41.7 (15.8) | 43.7 (14.4) | 46.0 (13.5) |
| | Unknown/Missing | 0 | 2 | 0 | 0 |
| Ethnicity | Asian | 88 (5.4%) | 43 (4.9%) | 63 (5.7%) | - |
| - n (%) | Black | 419 (25.6%) | 175 (19.8%) | 324 (29.3%) | - |
| | Mixed | 70 (4.3%) | 35 (4.0%) | 65 (5.9%) | - |
| | White | 988 (60.4%) | 492 (55.8%) | 593 (53.7%) | 499 (81%) |
| | Unknown/Missing | 70 (4.3%) | 137 (15.5%) | 60 (5.4%) | - |
| Marital status‡ | Divorced/Separated/Widowed | 211 (12.9%) | 91 (10.3%) | 108 (9.8%) | - |
| - n (%) | Married/Civil partner | 86 (5.3%) | 48 (5.4%) | 44 (4.0%) | - |
| | Single | 1311 (80.2%) | 619 (70.2%) | 927 (83.9%) | 406 (66%)** |
| | Unknown/Missing | 27 (1.7%) | 124 (14.1%) | 26 (2.4%) | - |
| Diagnosis§ | Dementia/organic disorder | 92 (5.6%) | 23 (2.6%) | 21 (1.9%) | - |
| - n (%) | Alcohol/substance misuse§§ | 86 (5.3%) | 92 (10.4%) | 63 (5.7%) | - |
| | Schizophrenia/psychosis | 921 (56.3%) | 344 (39.0%) | 703 (63.6%) | 381 (62%)*** |
| | Affective disorder | 209 (12.8%) | 113 (12.8%) | 120 (10.9%) | 169 (27%)**** |
| | Personality disorder | 139 (8.5%) | 77 (8.7%) | 71 (6.4%) | - |
| | Other | 51 (3.1%) | 18 (2.0%) | 29 (2.6%) | 66 (11%) |
| | Unknown/Missing | 137 (8.4%) | 215 (24.4%) | 98 (8.9%) | 3 (0.5%) |

*National survey of supported accommodation services in England 2014; 159 residential care service users, 251 supported housing and 209 floating outreach [1].

†Calculated from the median date within the search parameters (01-January-2008 to 31-December-2017, median date: 31-December-2012) and date of birth.

**'Never married or cohabited'.

***'Schizophrenia' & 'Schizoaffective disorder.

****'Bipolar affective disorder' & 'Depression or anxiety'.

‡The most frequently recorded marital status for individuals.

§The most recently recorded diagnosis.

§§Mental health or behavioural problem due to alcohol/substance misuse.

structured field to record the individual is residing in a supported accommodation service if this was not the case but there was a high level of missing data in the structured fields and human error is always a possibility, therefore using this approach alone could introduce bias. The free text search did facilitate a focus on a specific type of supported accommodation but this was only possible with knowledge of the local supported housing services, and using this approach will always return some false positives. The pros and cons of the two approaches therefore need to be weighed in deciding which to use or whether both are required. This will obviously depend on the focus of the evaluation being undertaken. For example, whilst the structured field search does not allow comparison of outcomes for different types of supported accommodation, combined with other routinely recorded clinical information (such as inpatient service use), it could potentially be used to evaluate the effectiveness of supported accommodation services. Free text data could be used to categorise the type of supported accommodation used for comparison, and to provide additional outcome data, such as whether the individual moved on to more independent accommodation successfully.

## Strengths and limitations

Most NHS Trusts have a clinical records policy that emphasises the importance of staff ensuring that certain fields are kept up to date to facilitate best practice and patient safety. However, the structured fields for accommodation used in this study had poor completion rates, an issue that has been well documented [12] and reported in other studies [18]. It is unknown, but as these variables are not completed by staff systematically, there may be reasons why these fields are not completed for some individuals whilst completed for others (e.g. greater stability in housing) which would lead to selection bias in this approach. There may also be causes for selection bias with the free text search approach as individuals with greater clinical contact are more likely to have a greater number of records and therefore more likely to be returned by the search. However, most people in supported accommodation have complex and longer term mental health problems [1] and are therefore likely to have an extensive history of contact with NHS mental health services. Poor completion rates are a limitation applicable to all research that uses routinely recorded data. A further issue relevant to all secondary research is that the data are not collected specifically for the purpose of the study and the potential research questions that can be addressed are limited by the available data. This ought to be balanced against the access to large datasets and the relatively low resource required. It should also be noted that there was a relatively good completion rate of the fields used in this study to collect sociodemographic data, including diagnosis. Also, we expect that as staff become more familiar with the technology and record systems, the accuracy of records and completion rates improves. However, to our knowledge this has not yet been confirmed by research but stratifying results by year in future validation studies may be of value.

Although we consider it a strength of the study that the researcher was able to complete the search in a short timeframe, more researcher time to investigate the availability and completion rate of further relevant variables would have strengthened the study. More time would also have allowed us to expand the free text searches to all uploaded documents and to all types of supported accommodation service. Also, using a second rater to manually review and code clinical notes would have increased the validity of our free text search of clinical notes.

As expected, developing the free text search took the majority of the time available for the study. However, an unforeseen issue arose that inevitably reduced the number of free text results. Service names, our key search terms, often also included part of the service address. As the patient's address is considered a Patient Identifier, all mentions of it in their records is either removed (in structured fields) or masked (in free text fields). Therefore, any patient using a service where the service name included any part of their address would not have been included in our free text search of clinical notes.

Finally, as our free text search approach requires knowledge of the local services available, contact with key personnel or use of a freedom of information request is needed.

## Future directions

An alternative approach would be to match the address field of patients to a list of known supported accommodation addresses within the Trust catchment area. The address field is a Patient Identifier so would need to be kept blinded to the researcher. Such an approach has already been used to identify admissions to care homes for older people in a study using the South London and Maudsley CRIS de-identified database [19].

Another possibility would be the use of natural language processing (NLP) to identify instances of supported accommodation in free text, or other potential clinical outcomes (e.g. successfully moving from a supported accommodation service to a more independent setting). NLP falls within the field of machine learning and is a technique used to autonomise the

process of analysing free text. It requires two data sets, a training set and test set. Both sets contain the same type of input data but only the training set contains the output. The output is typically generated by a human analysing the input data and then deciding what the output should be (similar to our manual notes review). The computer program analyses the training data set, looking for correlations between the input and output data, and creates an algorithm that can be used on the test data set to predict the output. This methodology has been widely and effectively applied to EHR systems [11]; many NLP applications have been developed to assist in the identification of samples/outcomes that are not readily available in structured fields (e.g. identifying symptoms of severe mental illness [20] and suicide ideation [21]).

## Conclusions

This study demonstrates it is possible to use de-identified EHR to identify a large sample of individuals who have used mental health supported accommodation services. This is a promising development in a field which is difficult and expensive to study through traditional research methods. However, it is important to consider the limitations of secondary research. Studies need to be designed with knowledge of the clinical data that is routinely collected, the variables that have sufficient completion rates and, for free text searches, the local supported accommodation services.

## Supporting information

**S1 Table. Original values in structured fields and their grouping term.**
(DOCX)

**S2 Table. Search results for the CPA structured field.**
(DOCX)

**S3 Table. Search results for the demographics structured field.**
(DOCX)

## Acknowledgments

The authors would like to acknowledge the support of the CIFT research database administration team and the UCLH NIHR Biomedical Research Centre.

## Author Contributions

**Conceptualization:** Christian Dalton-Locke, David Osborn, Helen Killaspy.

**Data curation:** Christian Dalton-Locke.

**Investigation:** Christian Dalton-Locke.

**Methodology:** Christian Dalton-Locke.

**Project administration:** Christian Dalton-Locke.

**Supervision:** Johan H. Thygesen, Nomi Werbeloff, David Osborn, Helen Killaspy.

**Writing – original draft:** Christian Dalton-Locke.

**Writing – review & editing:** Johan H. Thygesen, Nomi Werbeloff, David Osborn, Helen Killaspy.

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
