## [Decision Letter · Decision Letter 0]

19 May 2020

PONE-D-20-10079

Using de-identified electronic health records to research mental health supported housing services: a feasibility study

PLOS ONE

Dear Mr Dalton-Locke,

Thank you for submitting your manuscript to PLOS ONE. After careful consideration, we feel that it has merit but does not fully meet PLOS ONE’s publication criteria as it currently stands. Therefore, we invite you to submit a revised version of the manuscript that addresses the points raised during the review process.

We would appreciate receiving your revised manuscript by Jul 03 2020 11:59PM. To enhance the reproducibility of your results, we recommend that if applicable you deposit your laboratory protocols in protocols.io, where a protocol can be assigned its own identifier (DOI) such that it can be cited independently in the future. For instructions see: http://journals.plos.org/plosone/s/submission-guidelines#loc-laboratory-protocols

We look forward to receiving your revised manuscript.

Kind regards,

Sreeram V. Ramagopalan

Academic Editor

PLOS ONE

Journal Requirements:

Additional Editor Comments (if provided):

Reviewers' comments:

Reviewer's Responses to Questions

**Comments to the Author**

1. Is the manuscript technically sound, and do the data support the conclusions?

Reviewer #1: Partly

2. Has the statistical analysis been performed appropriately and rigorously? 

Reviewer #1: Yes

3. Have the authors made all data underlying the findings in their manuscript fully available?

Reviewer #1: No

4. Is the manuscript presented in an intelligible fashion and written in standard English?

Reviewer #1: Yes

5. Review Comments to the Author

Reviewer #1: Dear Editor,

Thank you very much for giving me the opportunity to review the manuscript: PONE-D-20-10079. In this feasibility study the authors investigated whether de-identified electronic health record (EHR) can be used effectively as a tool to identify large samples of users of mental health supported housing using structured fields and free text searches. The authors concluded that it is feasible and resource efficient to use the Clinical Record Interactive Search (CRIS) tool to identify individuals who have used mental health supported accommodation services.

The manuscript is well structured and is relevant in the digital world where we have data from patients with huge potential for research. The study is a good first step in utilizing EHR data in the field of mental health support services. The study sample is very large, and the authors discussed multiple ways to identify individuals with mental health supported accommodation services. However, there is potential to improve the description of methodology of the study and to add more analysis to increase the value of the study.

I see main problem with the methodology of the study that needs further clarification and explanation to convey the results of the study clearly.

First, this feasibility study explains different approaches to identify users of mental health supported accommodation services. But estimated true positive value of only one approach (i.e. free text search approach) is presented. It would add value to the manuscript if true positive values of the structured field search approaches are also estimated and presented. In line 265 the authors mentioned that it is unlikely that a clinician would add false information on supported accommodation service in structured field; however, there has been multiple validation studies showing less than perfect positive predictive value (PPV) of clinical diagnosis in EHR. Therefore, I think it is likely that the PPV of the CPA structured approach is less than 100%.

Second, the study is missing the ‘validity’ of the identification methods used by the authors. Ideally, manual chart review of ‘random’ samples from the identified individuals should be performed to get the positive predictive value (PPV) e.g. PPV of combining the CPA structured field approach and free-text approach; PPV of combining CPA structured approach and structured demographic field approach; and PPV of combining all three approaches together. The closest estimate provided is the true positive rate of free text search, which was performed for the first 10% of individuals after sorting the results by note date (line 166) (i.e. not random).

Having the information about the validity of the individual and combined search approach will certainly add value to this study.

Finally, the technical details in the Method section should be expanded to ensure that readers understand exactly how the authors identified individuals; and missingness and selection bias need to be further discussed.

Please see my detailed comments for each section below:

Abstract:

• The abstract is well-written.

• Line 28: the study is not based on data over the last 10 years.

• It would be informative if the authors can add something about the ‘setting’ of the study or add name of the mental health trust in the Method section.

• Result section line 34: “A manual review of these notes…” Please add, “…manual review of 10% of the notes…”.

• Result section, line 35-36: is there any reason of using the term ‘true positive rate’? I think more widely used term is ‘Positive Predictive Value’.

• Result section, line 39: The statement that these 337 individuals are likely to be false positive assumes that individuals identified by structured field search are all true positive. This is a strong assumption. Please see my comment on the Discussion section below.

• Conclusions: The term ‘efficient’ is very subjective and I suggest using it carefully. In this study authors have fixed the resources before the study. Hence, I cannot see the conclusion of efficiency is based on evidence generated by this study. Please see my comment on the Discussion section below.

• Conclusions are based only on results of structured fields, why free text is not mentioned?

Background:

• This section provides background and good overview of the key literature. However, the section is missing the background on the need of the problem addressed. It appears that the problem is ‘identification’ of people in EHR who have used mental health supported accommodation services. What methodologies have been used in the past to identify such people in the EHR in the same field or other closely related field using CRIS platform? And what were the challenges?

Methods:

Setting:

• Line 127-128: Are there any studies that have investigated the completion of EHR data over the years since it started in 2008? I expect the completion of data to improve over time. If there are differences, then it is a good idea to stratify the results according to years.

• I assume there must be changes to the EHR system or healthcare system in the 10 years of the study. Was there any reason of including all available EHR data since 2008 and not restricting the study to recent few years only? I assume the reason was to increase the sample size, but since it is a feasibility study a smaller sample would be acceptable.

Search approach:

• Line 135-136: “….sample in terms of their sociodemographics using structured fields, and…” I think Table 2 also has information from free text search.

• Line 136-137: What was the reason for deciding to compare the sociodemographic data to the national survey from 2014? The study is based on only 2 of the 326 local authority areas, spread from 2008-20017, and is known to be different (as the authors mentioned in the first paragraph of the Setting).

• Line 140-142: I could not find any details on how the authors will assess resource effectiveness? How it was measured and what was measured? We cannot assess any effectiveness by fixing the time (=resources). E.g. if we provide only 8 hours to work on something then we will get results, but quality will be compromised. Therefore, it depends on what quality was desired, which is not explained. Hence, we cannot make the resource assessment. However, I do agree that database studies are in general less resource demanding than a prospective real-world study or a clinical trial, but this study does not provide evidence supporting that.

Free text search of de-identified clinical notes:

• Line 166: “Results were ordered by identification number and note date.” What was the reason for sorting the results by date and the identification number? Ideally manual chart review should be on a random sample.

• Line 169: I assume true positive rate is same as positive predictive value (PPV). If this assumption is correct then the definition of true positive rate is incorrect, it should be the ratio of true positive to total positive. Although the calculation is correct in the result section, but the definition is incorrect.

• Line 186-187: Do the authors have any reference to support this methodology of balancing sensitivity and specificity? Any previous study that has used similar approach and calculated sensitivity and specificity providing evidence that this approach truly balances sensitivity & specificity?

Results

Structured fields search:

• What was the total base population? I assume it was 126,769 (line 128).

• Line 197-199: It is not clear how many individuals have no records of mental health accommodation services. I assume out of total individuals (i.e. 126,769), 1635 had records of mental health supported accommodation services, 9545 had missing/unknown values, and the rest did not get any mental health supported accommodation services. Is that correct? Please clarify. Also, it is not clear how many individuals in total had CPA field records.

• Missingness could reflect true absence of the of use of mental health supported accommodation. What was the assumption made for missing/unknown subjects? were they assumed to have no mental health support accommodation, or the data was ‘missing’? May be a flowchart would help.

• Table 1: The authors should add footnote explaining that the true positive value was derived from manual review of 10% of the identified individuals.

• Line 218-220: Please mention this exclusion criteria in the Method section, it is not currently explained there. Please also add how many individuals were excluded with this exclusion criteria in the final search.

• Line 218-220: “Therefore, a condition was added to the search whereby individuals were removed from the results if they only had a single note matching the search term.” why these individuals were not added again after iterating the search term?

Comparing the structured fields and free text search approach:

• Table 2: It does not illustrate only the sociodemographic but also clinical characteristics.

Discussion:

• Line 255-261: These can be moved to result section or deleted.

• Line 260-261: “….it is likely that many of these 337 are false positives.” This is a big assumption considering we have ‘estimated’ true positive rates. With similar assumptions 45.2% (Figure 2) individuals identified in CPA structured field would ‘likely’ be false positive as they appeared only in CPA structured search. Because the authors did not identify exactly who are false positive in text search field and the authors did not estimate true positive value for CPA structured field search, it will be difficult to conclude anything.

• Line 285, “However, the free text search did not appear to significantly enhance sensitivity.” The authors did not estimate ‘sensitivity’ in this study, so there is no data to support this statement.

• Line 299: “However, an unforeseen issue arose that inevitably reduced the number of free text results”. I think it would be worthwhile to mention what was the issue and how it had impacted the results. If it was fixed, then this can be deleted.

• May be the authors can put light on impact of ever-changing technology during the study. I assume there has been changes to EHR and I assume quality of data (e.g. completion) vary as familiarity to the system increase.

• Some of the single service searches have reached PPV of 100% and others remained as low as 35%. Is it possible that the data quality varies between different housing services? Is it possible that data quality varies between providers as well? e.g. some clinicians would not ask or report on accommodation status?

• I assume the structured variable that are used in this study to identify the cases are not systematically reported. Therefore, missingness is not at random leading to selection bias. Maybe it is worthwhile to discuss selection bias in this study.

Conclusions:

• I think the authors’ claim of ‘resource efficiency’ is not substantially supported with evidence.

Other comments:

• References are not as per the journal’s acceptable style. Many are missing the volume and page number or DOI.

• The authors have used some terms inconsistently e.g. effectiveness and efficiency are interchangeable. I suggest using terms consistently to make the manuscript easy to read.

6. PLOS authors have the option to publish the peer review history of their article (what does this mean?). If published, this will include your full peer review and any attached files.

Reviewer #1: No

---

## [Author Response · Author response to Decision Letter 0]

29 Jul 2020

UCL DIVISION OF PSYCHIATRY

FACULTY OF BRAIN SCIENCES

23 July 2020

Dear Sreeram,

Many thanks to you and the reviewer for the very helpful comments and opportunity to improve our manuscript. 

As requested, we have responded to the reviewer’s comments (in red, below) and have submitted a version of the manuscript with tracked changed to illustrate the revisions made since the original submission, as well as a clean version without tracked changes. 

We hope that you both find this satisfactory and look forward to learning your response.

Yours sincerely

Christian Dalton-Locke

PhD student

Journal Requirements:

Files renamed and separate files created for supporting documents (previously ‘Appendices’). 

There are ethical and legal restrictions on sharing this data set. The data cannot be shared upon request unless the person is an authorised researcher adhering to the relevant protocols. Further details to this can be found under the headings: ‘Ethics approval and consent to participate’ and ‘Availability of data and material’.

Additional Editor Comments (if provided):

Reviewers' comments:

Reviewer's Responses to Questions

Comments to the Author

1. Is the manuscript technically sound, and do the data support the conclusions?

Reviewer #1: Partly

2. Has the statistical analysis been performed appropriately and rigorously? 

Reviewer #1: Yes

3. Have the authors made all data underlying the findings in their manuscript fully available?

Reviewer #1: No

4. Is the manuscript presented in an intelligible fashion and written in standard English?

Reviewer #1: Yes

5. Review Comments to the Author

Reviewer #1: Dear Editor,

Thank you very much for giving me the opportunity to review the manuscript: PONE-D-20-10079. In this feasibility study the authors investigated whether de-identified electronic health record (EHR) can be used effectively as a tool to identify large samples of users of mental health supported housing using structured fields and free text searches. The authors concluded that it is feasible and resource efficient to use the Clinical Record Interactive Search (CRIS) tool to identify individuals who have used mental health supported accommodation services.

The manuscript is well structured and is relevant in the digital world where we have data from patients with huge potential for research. The study is a good first step in utilizing EHR data in the field of mental health support services. The study sample is very large, and the authors discussed multiple ways to identify individuals with mental health supported accommodation services. However, there is potential to improve the description of methodology of the study and to add more analysis to increase the value of the study.

I see main problem with the methodology of the study that needs further clarification and explanation to convey the results of the study clearly.

First, this feasibility study explains different approaches to identify users of mental health supported accommodation services. But estimated true positive value of only one approach (i.e. free text search approach) is presented. It would add value to the manuscript if true positive values of the structured field search approaches are also estimated and presented. 

Thank you for this comment and thoughtful suggestion. Estimated true positive values of the structured field approaches would indeed add value to this study but unfortunately we do not believe it feasible to obtain such a measure. The closest we would be able to get to an estimate of the true positive values of the structured field approach would be to review uploaded documents and/or clinical notes that are recorded around the same date the structured field is completed. However, we believe that this would likely produce an estimate that greatly underestimates the actual true positive rate from this sample. This is mainly due to the fact that it is not standard or routine practice for NHS staff to record whether their patients are using supported accommodation or not (these services are not usually provide by the NHS), and because they have recorded it using structured fields doesn’t necessarily mean they will also record it in clinical notes/ uploaded documents. Furthermore, the de-identified nature of the data means it is not possible to use data available outside of the EHR system to validate the structured fields.

In line 265 the authors mentioned that it is unlikely that a clinician would add false information on supported accommodation service in structured field; however, there has been multiple validation studies showing less than perfect positive predictive value (PPV) of clinical diagnosis in EHR. Therefore, I think it is likely that the PPV of the CPA structured approach is less than 100%.

We agree that structured fields on accommodation status are unlikely to be 100% accurate, human error is always going to be an issue/limitation when using health records for research – we have made a tracked change to reflect this (p.19). 

Second, the study is missing the ‘validity’ of the identification methods used by the authors. Ideally, manual chart review of ‘random’ samples from the identified individuals should be performed to get the positive predictive value (PPV) e.g. PPV of combining the CPA structured field approach and free-text approach; PPV of combining CPA structured approach and structured demographic field approach; and PPV of combining all three approaches together. 

As described above, we agree that adding PPV for the structured fields would add value to this study, but unfortunately we believe it unfeasible to obtain accurate estimates of this. We feel that showing the overlap between the three different approaches still provides a degree of validation of each approach (Figure 2, Venn diagram, p.16) – a researcher would reasonably be more confident of their sample using the overlap of all three indicators rather than working on a sample identified from just one of the approaches.

The closest estimate provided is the true positive rate of free text search, which was performed for the first 10% of individuals after sorting the results by note date (line 166) (i.e. not random).

Thank you for this comment. The results were ordered by patient identification number, which is randomly assigned by CRIS, and then by note date, therefore the sample is random but this is not clear. We have added tracked changes to clarify this (p.9).

Having the information about the validity of the individual and combined search approach will certainly add value to this study.

Finally, the technical details in the Method section should be expanded to ensure that readers understand exactly how the authors identified individuals; and missingness and selection bias need to be further discussed.

I have added text explaining exactly which values in the structured fields were used to identify our sample (p.8-9), and a reference to table S1 which lists all possible response options for both fields. Have also added discussion on potential selection bias of both search approaches under ‘Limitations’ (p.19-20).

Please see my detailed comments for each section below:

Abstract:

• The abstract is well-written.

• Line 28: the study is not based on data over the last 10 years.

Thank you for spotting this. This line has been corrected (p.2).

• It would be informative if the authors can add something about the ‘setting’ of the study or add name of the mental health trust in the Method section.

Name of mental health Trust added (p.2).

• Result section line 34: “A manual review of these notes…” Please add, “…manual review of 10% of the notes…”.

Added (p.2).

• Result section, line 35-36: is there any reason of using the term ‘true positive rate’? I think more widely used term is ‘Positive Predictive Value’.

We agree, thank you for this comment. All mentions of ‘true positive rate’ changed to ‘positive predictive value’ (p.2).

• Result section, line 39: The statement that these 337 individuals are likely to be false positive assumes that individuals identified by structured field search are all true positive. This is a strong assumption. Please see my comment on the Discussion section below.

We feel this statement only assumes individuals appearing in the free text search and also in one of the structured fields is more likely to be a true positive than those appearing only in the free text search, which we think is a reasonable assumption.

• Conclusions: The term ‘efficient’ is very subjective and I suggest using it carefully. In this study authors have fixed the resources before the study. Hence, I cannot see the conclusion of efficiency is based on evidence generated by this study. Please see my comment on the Discussion section below.

The word ‘efficient’ replaced with ‘requires minimal resources’ (p.3).

• Conclusions are based only on results of structured fields, why free text is not mentioned?

Mention of free text approach added (p.3).

Background:

• This section provides background and good overview of the key literature. However, the section is missing the background on the need of the problem addressed. It appears that the problem is ‘identification’ of people in EHR who have used mental health supported accommodation services. What methodologies have been used in the past to identify such people in the EHR in the same field or other closely related field using CRIS platform? And what were the challenges?

Thank you for this comment, enabling us to address this. We have added text to the Background (p.6) explaining de-identified EHR have not yet been used to research mental health supported accommodation. Regarding challenges, we have added that although supported accommodation services are not usually provided by the NHS, there is potential for using CRIS to identify them as most are also using NHS mental health services.

Methods:

Setting:

• Line 127-128: Are there any studies that have investigated the completion of EHR data over the years since it started in 2008? I expect the completion of data to improve over time. If there are differences, then it is a good idea to stratify the results according to years.

We found a study reviewing the evolution of EHR since 1992 (https://www.ncbi.nlm.nih.gov/pmc/articles/PMC5171496/) but we are unaware of any study reporting completion rates of records by year. However, we do agree it is likely that as staff become more familiar with the records system and the technology, completion rates improve, and so have added text discussing this (p.20).

• I assume there must be changes to the EHR system or healthcare system in the 10 years of the study. Was there any reason of including all available EHR data since 2008 and not restricting the study to recent few years only? I assume the reason was to increase the sample size, but since it is a feasibility study a smaller sample would be acceptable.

We wanted to see if we could identify a relatively large sample, not just a sample, therefore we wanted to maximise our sampling frame and the period of time looked at, especially given that lengths of stay at these services are four years on average (https://bmcpsychiatry.biomedcentral.com/articles/10.1186/s12888-016-0797-6). 

Search approach:

• Line 135-136: “….sample in terms of their sociodemographics using structured fields, and…” I think Table 2 also has information from free text search.

Here, we meant that we used structured fields to investigate the sociodemographics (and clinical characteristics) of the sample identified using the different search approaches (including the free text search), and compare this to a national survey. We have added text here to hopefully make this clear (p.18).

• Line 136-137: What was the reason for deciding to compare the sociodemographic data to the national survey from 2014? The study is based on only 2 of the 326 local authority areas, spread from 2008-20017, and is known to be different (as the authors mentioned in the first paragraph of the Setting).

This national survey provides the best description available of individuals who use mental health supported accommodation services, and so although different, we think it still a useful comparison to make. The last sentence in this section notes this difference and its effect on the results in this table (p.18).

• Line 140-142: I could not find any details on how the authors will assess resource effectiveness? How it was measured and what was measured? We cannot assess any effectiveness by fixing the time (=resources). E.g. if we provide only 8 hours to work on something then we will get results, but quality will be compromised. Therefore, it depends on what quality was desired, which is not explained. Hence, we cannot make the resource assessment. However, I do agree that database studies are in general less resource demanding than a prospective real-world study or a clinical trial, but this study does not provide evidence supporting that.

We wanted to show whether this approach could return a large sample with limited researcher hours compared to what would normally be invested in a primary research study requiring participant recruitment and research interviews. We have added text to the Methods to clarify this (p.8). 

Free text search of de-identified clinical notes:

• Line 166: “Results were ordered by identification number and note date.” What was the reason for sorting the results by date and the identification number? Ideally manual chart review should be on a random sample.

Thank you for noting this. As described above, text has been added to clarify how this is a random sample (p.9).

• Line 169: I assume true positive rate is same as positive predictive value (PPV). If this assumption is correct then the definition of true positive rate is incorrect, it should be the ratio of true positive to total positive. Although the calculation is correct in the result section, but the definition is incorrect.

Thank you for noting this. Definition corrected (p.9).

• Line 186-187: Do the authors have any reference to support this methodology of balancing sensitivity and specificity? Any previous study that has used similar approach and calculated sensitivity and specificity providing evidence that this approach truly balances sensitivity & specificity?

Have added text to explain the reason for this methodology: iterations were an attempt to increase/maintain accuracy whilst maintaining number of returns (p.10).

Results

Structured fields search:

• What was the total base population? I assume it was 126,769 (line 128).

• Line 197-199: It is not clear how many individuals have no records of mental health accommodation services. I assume out of total individuals (i.e. 126,769), 1635 had records of mental health supported accommodation services, 9545 had missing/unknown values, and the rest did not get any mental health supported accommodation services. Is that correct? Please clarify. Also, it is not clear how many individuals in total had CPA field records.

• Missingness could reflect true absence of the of use of mental health supported accommodation. What was the assumption made for missing/unknown subjects? were they assumed to have no mental health support accommodation, or the data was ‘missing’? May be a flowchart would help.

This section has largely been re-written to clarify the issues raised here (p.10).

• Table 1: The authors should add footnote explaining that the true positive value was derived from manual review of 10% of the identified individuals.

Footnote to table added (p.14).

• Line 218-220: Please mention this exclusion criteria in the Method section, it is not currently explained there.

Have added following text under Methods sub-section Free text search (p.10): “Patterns to false positives were not limited to text included in the clinical note but could also include patterns such as the number of notes returned per individual. For example, if a false positive were more likely than a true positive to only have a single note returned by the search, then the search could be refined to only include individuals which have more than one note pertaining to them.”

 Please also add how many individuals were excluded with this exclusion criteria in the final search.

Have added text stating difference in number of individuals and notes for first search and first iteration (p.15).

• Line 218-220: “Therefore, a condition was added to the search whereby individuals were removed from the results if they only had a single note matching the search term.” why these individuals were not added again after iterating the search term?

Because it was reasoned that individuals with single notes returning would still more likely (but not always) be false positives. The logic behind this was that if an individual really was using supported accommodation then they would likely have more than one clinical note in reference to this, and so more than one note returned by the search.

Comparing the structured fields and free text search approach:

• Table 2: It does not illustrate only the sociodemographic but also clinical characteristics.

Table title and relevant text in main text changed to reflect this (p.17-18).

Discussion:

• Line 255-261: These can be moved to result section or deleted.

Deleted (p.18).

• Line 260-261: “….it is likely that many of these 337 are false positives.” This is a big assumption considering we have ‘estimated’ true positive rates. With similar assumptions 45.2% (Figure 2) individuals identified in CPA structured field would ‘likely’ be false positive as they appeared only in CPA structured search. Because the authors did not identify exactly who are false positive in text search field and the authors did not estimate true positive value for CPA structured field search, it will be difficult to conclude anything.

Deleted (p.18).

• Line 285, “However, the free text search did not appear to significantly enhance sensitivity.” The authors did not estimate ‘sensitivity’ in this study, so there is no data to support this statement.

Correct, thank you for spotting this. This text has been deleted (p.20).

• Line 299: “However, an unforeseen issue arose that inevitably reduced the number of free text results”. I think it would be worthwhile to mention what was the issue and how it had impacted the results. If it was fixed, then this can be deleted.

The issue is explained in the text following this line, and due to the nature of this issue, it was not possible to calculate the impact of this nor were we able to fix it (p.20-21).

• May be the authors can put light on impact of ever-changing technology during the study. I assume there has been changes to EHR and I assume quality of data (e.g. completion) vary as familiarity to the system increase. Potential health inequalities and disparities exacerbated by interventions put in place to reduce transmission of COVID-19, and by reduced access to health and social care services.

We agree the impact of change of technology and general public familiarity with it (including healthcare clinicians) on completion and quality of EHR is interesting and worth researching, we feel that is it perhaps beyond the scope of this study to address it here but have added text to suggest as a useful addition in future studies (p.20).

• Some of the single service searches have reached PPV of 100% and others remained as low as 35%. Is it possible that the data quality varies between different housing services? Is it possible that data quality varies between providers as well? e.g. some clinicians would not ask or report on accommodation status?

It is possible, but we believe it is more likely due to the varying distinctiveness of the names of services. The free text search was based on the service name, so a more distinctive service name would produce a search with less false positives.

• I assume the structured variable that are used in this study to identify the cases are not systematically reported. Therefore, missingness is not at random leading to selection bias. Maybe it is worthwhile to discuss selection bias in this study.

Text added to Strengths and limitations (p.19-20) regarding potential selection bias:

“It is unknown, but as these variables are not completed by staff systematically, there may be reasons why these fields are not completed for some individuals whilst completed for others (e.g. greater stability in housing) which would lead to selection bias in this approach. There may also be causes for selection bias with the free text search approach as individuals with greater clinical contact are more likely to have a greater number of records and therefore more likely to be returned by the search. However, most people in supported accommodation have complex and longer term mental health problems (1) and are therefore likely to have an extensive history of contact with NHS mental health services.”

Conclusions:

• I think the authors’ claim of ‘resource efficiency’ is not substantially supported with evidence.

Agreed, claim deleted (p.22). 

Other comments:

• References are not as per the journal’s acceptable style. Many are missing the volume and page number or DOI.

Volume and page number or DOI added where missing. Unfortunately, these corrections do not appear as tracked changes because of the referencing software used (p.25-27).

• The authors have used some terms inconsistently e.g. effectiveness and efficiency are interchangeable. I suggest using terms consistently to make the manuscript easy to read.

Thank you for the noting this. Use of the terms ‘effectiveness’ and ‘efficiency’ have been reviewed throughout the manuscript and we believe are now consistently used, and not used interchangeably.

---

## [Editor Report · Decision Letter 1]

31 Jul 2020

Using de-identified electronic health records to research mental health supported housing services: a feasibility study

PONE-D-20-10079R1

Dear Dr. Dalton-Locke,

We’re pleased to inform you that your manuscript has been judged scientifically suitable for publication and will be formally accepted for publication once it meets all outstanding technical requirements.

Kind regards,

Sreeram V. Ramagopalan

Academic Editor

PLOS ONE
---

## [Editor Report · Acceptance letter]

11 Aug 2020

PONE-D-20-10079R1 

Using de-identified electronic health records to research mental health supported housing services: a feasibility study 

Dear Dr. Dalton-Locke:

I'm pleased to inform you that your manuscript has been deemed suitable for publication in PLOS ONE. Congratulations! Your manuscript is now with our production department. 

Kind regards, 

on behalf of

Dr. Sreeram V. Ramagopalan 

Academic Editor

PLOS ONE